# Assessing the Potential of Remotely-Sensed Drone Spectroscopy to Determine Live Coral Cover on Heron Reef

**Valerie J. Cornet [1,\*] and Karen E. Joyce [2]**

[1] College of Science and Engineering, James Cook University Bebegu Yumba Campus, Townsville, QLD 4811, Australia

[2] College of Science and Engineering/TropWATER, James Cook University Nguma-bada Campus, Cairns, QLD 4878, Australia; karen.joyce@jcu.edu.au

**\*** Correspondence: valerie.cornet@my.jcu.edu.au

**Abstract:** Coral reefs, as biologically diverse ecosystems, hold significant ecological and economic value. With increased threats imposed on them, it is increasingly important to monitor reef health by developing accessible methods to quantify coral cover. Discriminating between substrate types has previously been achieved with in situ spectroscopy but has not been tested using drones. In this study, we test the ability of using point-based drone spectroscopy to determine substrate cover through spectral unmixing on a portion of Heron Reef, Australia. A spectral mixture analysis was conducted to separate the components contributing to spectral signatures obtained across the reef. The pure spectra used to unmix measured data include live coral, algae, sand, and rock, obtained from a public spectral library. These were able to account for over 82% of the spectral mixing captured in each spectroscopy measurement, highlighting the benefits of using a public database. The unmixing results were then compared to a categorical classification on an overlapping mosaicked drone image but yielded inconclusive results due to challenges in co-registration. This study uniquely showcases the potential of using commercial-grade drones and point spectroscopy in mapping complex environments. This can pave the way for future research, by increasing access to repeatable, effective, and affordable technology.

**Keywords:** remote sensing; coral reefs; drones; linear unmixing; R; google earth engine

## 1. Introduction

Coral reefs are some of the most biologically diverse ecosystems on the planet, providing key ecosystem services to coastal communities through tourism, food security, and coastal protection [1]. However, reefs around the world are currently experiencing decline, through mass coral bleaching, ocean acidification, and water quality reduction [2]. Due to both their ecological and economic importance, more accessible and cost-effective methods to map and monitor the decline of coral reefs are needed.

Many monitoring programs have focused on studying reefs locally using in situ field methods [3]. Due to the various difficulties of working in aquatic environments, there is increasing pressure to develop better, wide-scale methods to map and monitor coral reef benthos [4,5]. Collecting data through using remote sensing therefore complements research conducted in the field. By developing more affordable and repeatable methods in remote sensing, research can be made more accessible and efficient. This is particularly useful in locations that are hard to access as the improved capacity to survey remote areas can facilitate repeated monitoring [6]. This can be achieved at broad spatial and temporal scales, using platforms such as drones, aircrafts, or satellites.

Drone-based remote sensing presents a wide array of advantages with regard to local, detailed assessments of study sites. With advances in the technological field over the years, the cost of using drone-mounted sensors has decreased, making consumer-grade drones accessible to many whilst reducing the need for expertise in operating commercial

grade drone technology [7]. Increased battery life has led to increased flight time, and decreased payload weight has made drones lighter and more user friendly [8]. Additionally, on-demand deployment has the advantage of choosing favourable weather conditions for collecting data [9]. Drones provide the benefit of flying under the cloud cover, resulting in greater flexibility in terms of data collection time frames compared to satellites and aircrafts. Furthermore, since external limitations presented by the environment influences the accuracy of benthic mapping studies, reducing the distance between the sensor and the subject reduces atmospheric effects on readings [10]. These combined benefits give drones competitive advantage over other remote sensing platforms.

However, there are disadvantages in using drones that need to be considered as well. Data processing errors often occur within the quantitative analysis and classification steps. As for errors in data collection, these are presented by the sensors and platforms, the classification steps, and the environment. When collecting data with drones, it is important to note that the platform moves. When doing so, attached spectrometers do not always point directly downwards. This means that spectral readings may not always be taken from the area of interest [11]. Errors in data collection from drones may also occur through geopositioning, as the accuracy of the GPS location determined by the drone is not always exact. This is particularly the case in commercial-grade drones, as the inertial navigation systems that measure position information are often of low to medium accuracy to save costs and payload weight [12]. Similar to the inaccuracies present by drones' GPS, in-water validation imagery collected in situ are also subject to spatial inaccuracy. This, along with scale differences in field data justify the difficulties to use field data for direct comparisons to aerial mapping [13].

Mapping and monitoring using remote sensing often relies on being able to accurately record colour or light interactions in the environment [14]. This includes using spectrometers to make measurements of reflection, absorption, and transmission, and finding patterns or 'spectral signatures' that may be unique to features of interest – in this case, live coral, algae, rock, and sand. This information can be collected using imaging spectrometers (e.g., hyperspectral scanners) or with individual point based spectroscopy [14]. Spectroscopy has been used to distinguish between live coral and other coral reef benthos in the past, but these studies have largely been limited to in situ underwater or close-range measurements [15–20]. Capturing data in that way is time intensive and provides limited coverage.

However, drone-based spectroscopy provides the opportunity to extend the coverage, providing a tool for rapid data collection. While other research has documented the potential for using small and lightweight imaging spectrometers on drone platforms (e.g., [21], little work has been done to test the extent to which the more affordable point-based spectrometers can also capture categorical and continuous variable information about the benthos and water column.

In using drones, a major consideration is the influence of the water column and the nature of its influence under different light and environmental conditions, such as waves. With varying depths and water quality, there is likely increased confusion between more spectrally similar classes such as algae and coral due to uneven attenuation throughout spectral signatures [18]. For example, it has been found that with higher chlorophyll or sediment content in the water, more algae will likely be classified as coral [22]. Lee et al. [23] proposed a widely used inversion model that uses diffuse attenuation coefficients as functions of light absorption and scattering. This model was built upon to derive water column properties and water depth, which has been widely used in water column correction [24,25]. Classification of benthic groups was successfully achieved by Goodman and Ustin [24] through combining Lee et al. [25]'s semi-analytical inversion model with linear spectral unmixing, which allowed for the correction of the water column and achieved an overall accuracy of 80% for all substrate groups. BRUCE, a model built upon Lee et al. [25]'s algorithm achieved an overall accuracy of 79% in mapping benthic substrates [26].

However, in clear, unturbid, and shallow waters under 5 metres, water column correction is not always necessary to capture accurate measurements of the benthos.

This study tests the extent to which consumer-grade drones are capable of providing fine resolution information on coral reefs. This type of data offers a low-cost resource that has the potential to overcome separability issues between classes such as coral and algae, as well as a level of detail and information that cannot be provided by multispectral and RGB data. By using drones, there is the potential to bridge the scale gaps presented between field and satellite-based assessments. Achieving this would help pave the way for future research in the field of remote sensing, as it would demonstrate how accessible technology such as consumer-grade drones and public spectral endmember libraries can be used by anyone. As such, the aim of this study is to quantify the amount of various benthic substrates using drone-based spectroscopy on Heron Reef.

## 2. Materials and Methods

### 2.1. Study Site

Data were collected at Heron Reef (23.44°S, 151.91°E), a shallow, lagoonal coral reef located on the Southern end of the Great Barrier Reef, Australia (Figure 1). The shallow depth of the reef and the clear water afforded by its offshore location allow for effective spectral data collection. As it is a lagoonal reef, the depth remains relatively constant across the reef.

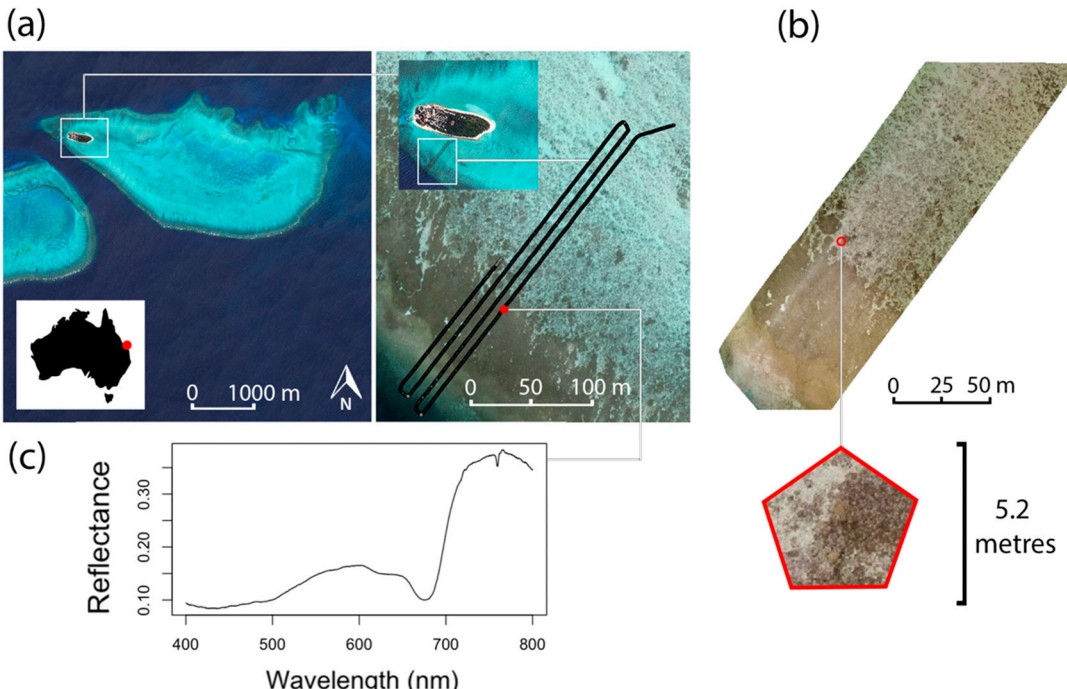

**Figure 1.** (**a**) Heron Reef study site. Image obtained from Google Earth and shows Heron Island, Heron Reef and the lagoon, along with the primary drone flight used to establish workflow. Drone flight is indicated as a series of black points. Satellite images were obtained from Google Earth. (**b**) Mosaicked RGB image of reef and study area on Google Earth Engine. Note that in Google Earth Engine the creation of completely circular areas is not possible and therefore the use of a pentagon was used to get the closest equivalence for linear unmixing results. (**c**) Example spectrum from drone flight sampled from the high coral cover region of the reef.

### 2.2. Data Collection

#### 2.2.1. Public Spectral Library

We used a spectral library of known features to calibrate and validate our drone spectroscopy mapping model. For the purpose of this study, we defined the benthic substrate features of interest (spectral endmembers) as live coral, algae, sand, and rock as these are

the most common generalised substrate groups found at the study site [27]. Representative spectra were chosen from the public spectral library of substrata collected in situ on Heron Island in 2006 by Dr Christian Roelfsema and Dr Stuart Phinn [28]. The public library consisted of endmember spectra that were recorded at shallow depths using a dive torch as a light source 5 cm away from the subject and a white panel was used as a baseline to calibrate the respective spectrometers. The dark current of the spectrometer (concurrent to the thermal variation) was also accounted for, removing the effects of dark current noise. Recordings of the digital number obtained were converted to reflectance values through the equation below, where dark current is written as Dark, Target refers to the reflectance of the target, and White refers to the reflectance of the white panel:

$$R = \frac{(Target - Dark)}{(White - Dark)} \tag{1}$$

### 2.2.2. Drone Spectroscopy

Spectroscopy data were collected using an Ocean Optics STS-Vis 15° field of view spectrometer, which measured reflectance at bands within the effective spectral range of 350 to 800 nm mounted on a 3DR Solo drone [29]. At a flying altitude of 20 m, this achieved an approximate spectral and spatial resolution of 0.13 nm and 5.2 m, respectively (Figure 2). Point spectroscopy data were collected approximately four times per second and each data point was attributed with the time and coordinates of the drone at the time of capture. The drone was flown up and down adjacent flight paths using a trajectory perpendicular to the shore in order to obtain a cross-reef-flat study area (Figure 1a). Data were calibrated to reflectance using a 99% Spectralon® reference panel (Labsphere) [22]. A Phantom 4 Pro, with an RGB camera also captured photos over the same region for accuracy assessment.

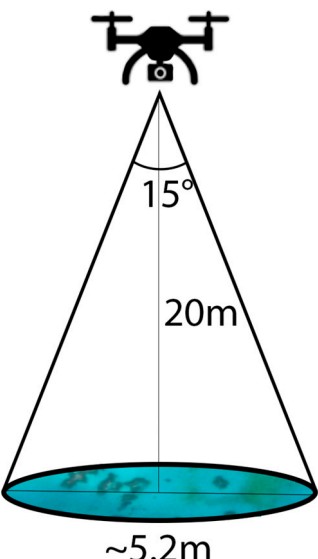

**Figure 2.** Drone footprint of flight path for spectrometer data collection. Since drone was flown at a constant height of 20 m, a point resolution of approximately 5.2 m diameter was achieved.

### 2.3. Data Processing

As seen in Figure 3, there are three subsequent steps in the methods tested which are described below. As it relies entirely on running the code written on R and Google Earth Engine, there is no need for expertise regarding commercial software, nor is there the need to obtain licenses for paid software (see Supplementary Materials).

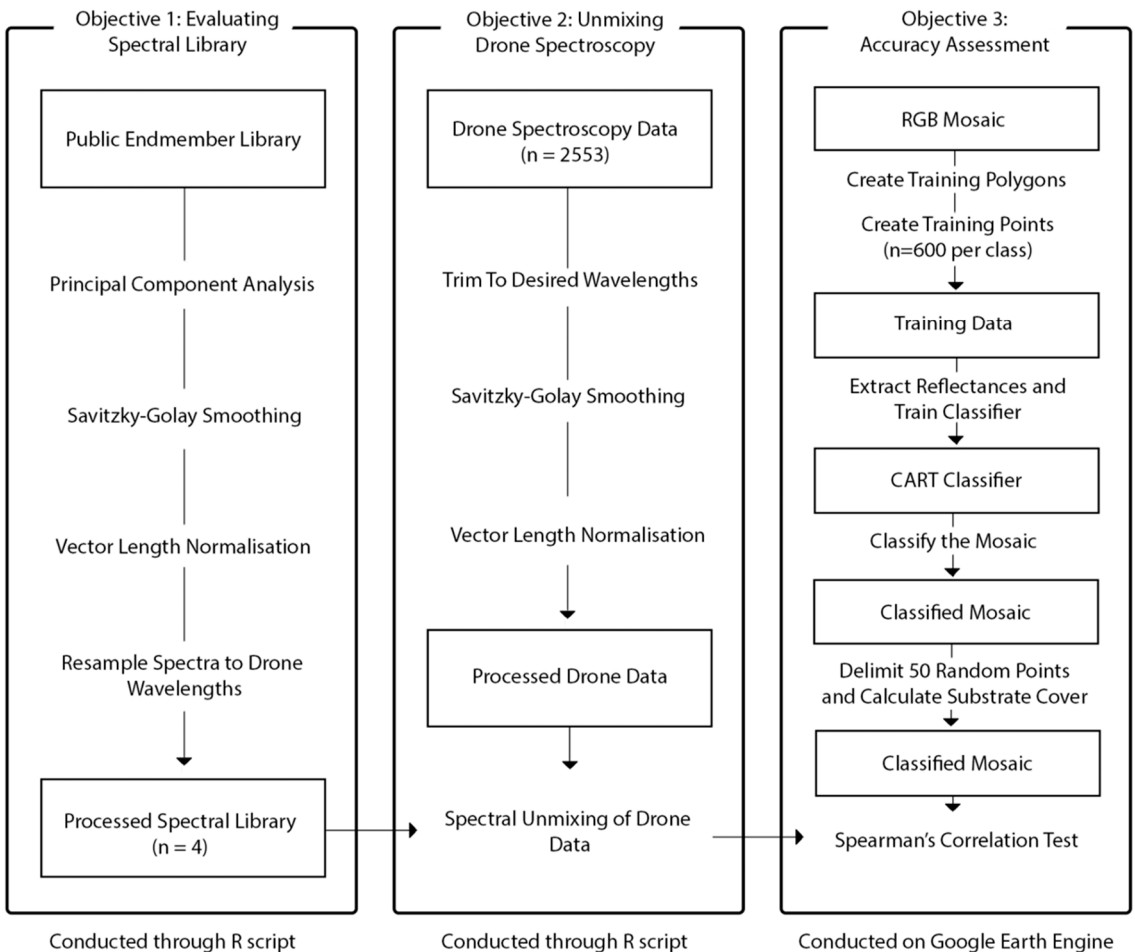

**Figure 3.** Workflow of the study. Each column represents the workflow to achieve the three objectives. The first column demonstrates the steps to obtain and modify the spectral library from a public database to obtain the four spectra for live coral, sand, algae, and rock. The second column shows the steps to obtain the processed drone data. These two datasets will be used in combination to derive the fractional contributions of each endmember class. The third column shows the steps of the accuracy assessment. The output datasets are shown in boxes and unboxed comments represent the steps for each objective.

### 2.3.1. Evaluating the Spectral Library

To choose endmember spectra, a principal component analysis (PCA) was conducted on all the endmembers present in the public library [30]. This method to visualise the maximum variation seen between data points has been used in studies proving its use in endmember determination [31]. Spectra that were projected far apart from other substrate classes and within their own substrate class were chosen. Preliminary review comparing spectral signatures to known "pure" endmember signatures was conducted to confirm suitability of the endmember.

Spectra were processed to create the final endmember library of the four substrate classes (Figure 3). Spectra were smoothed through Savitzy–Golay smoothing and normalised for vector length. Vector length normalisation involves calculating the length of reflectance vectors and dividing reflectance values by the vector length [32]. This ensures that a focus is given on the shape of the spectral signatures in the spectral unmixing step rather than the absolute values. Finally, the spectra were tested for collinearity using the detect.lindep()function in R from the plm package [33,34]. There should be no collinearity or linear dependence detected between endmember spectra as this is likely to lead to misclassification.

### 2.3.2. Evaluating Drone Spectroscopy

A separate endmember library consisting of endmembers sampled from the hyperspectral drone data were created through conducting a principal component analysis of the data and choosing the "purest" endmembers found clustered furthest apart. This was used as a comparison to the public endmember library in order to evaluate the pros and cons of each. Sampling within the studied dataset gives the advantage of providing spectra that will inherently be sourced from the same sensor and in the same environmental conditions. However, the likelihood of providing "pure" spectra is low due to the resolution of the spectroscopy data and the fine scale of spatial heterogeneity of the reef benthos.

Spectral reflectance values of the final endmembers were corrected for through smoothing and normalising in the same manner as the drone data, as explained above. Spectra were then resampled in order to coordinate with the wavelengths sampled in the drone spectroscopy dataset. Resampling was conducted through the resample() function in the spectrolab package using R [33,35]. The algorithm will then be used to separate these endmembers and determine the fractional contribution of each endmember. Through this, live coral cover may be estimated. This section of the workflow was processed in R (Figure 3) [33].

### 2.3.3. Unmixing Drone Spectroscopy

Spectra were imported into and processed in R in the appropriate format to run the code (columns as wavelengths and rows as individual points) and work through the steps of objective 2 in the workflow (Figure 3) [33]. Prior to the unmixing step, spectra from the drone data were also smoothed using Savitzy–Golay smoothing and normalised, through vector length normalisation [32,36]. Spectra collected by the drone were subset to record reflectance between 400 and 750 nm due to the opaque nature of the water column at wavelengths above and atmospheric scattering below that in the visible spectrum. Data reduction serves in reducing dimensionality of the dataset, which further facilitates algorithm performance, complexity, and data storage [37]. Due to the time limitations presented by the study and the aim of shaping a more accessible, repeatable, and relatively simple workflow, water column was not corrected for using radiative transfer equations. Previous studies have confirmed that classification of reef substrata using the aforementioned spectral range remains possible at depths shallower than six meters, which was the case for this study [38].

A single endmember spectral mixture analysis (SMA) was conducted to unmix endmembers for the hyperspectral data obtained. This was chosen because previous studies, have established its ability to unmix benthic classes, its accessibility of unmixing algorithms, and the lower computational power needed compared to MESMA or non-linear SMAs. Single endmember unmixing functions as a linear unmixing method. This assumes a linear contribution of endmembers to the spectra. This implies that the fractional spatial contribution of an endmember will equal the fractional spectral contribution an endmember will have on a spectrum. Although it is unlikely that the nature of spectral mixing among reef substrata is completely linear, most coral reef benthic studies that have used this method have yielded positive results [20,39]. The lack of perfect linearity in coral reef systems could be explained by the morphologic nature of coral colonies, where spectral reflectance may differ depending on the viewing angle of the colony [40]. This is also important when considering different substrate types overlaying one another. For example, a coral colony may have a dead top that might present as turf algae, whilst the rest of the coral colony below classifies as live coral. Despite this, using a linear unmixing model provides the additional advantage of being less sensitive to collinearity between endmembers [41]. This is useful for this study as live coral spectra and algae spectra are known to be highly similar, resulting in an increased likelihood of estimation errors if a non-linear model is used.

Non-negative and least squares (NNLS) constraints were applied to carry out simultaneous inversion of the data and endmember determination. The inversion step allows the fractional abundances retrieved to be constrained to be non-negative, meaning that all

fractions within a pixel will be positive, rendering the results more realistic over unconstrained methods. The model was not forced to sum to one, to give a better indication of the unexplained spectral contributions by endmembers. If the summed fractional contributions obtained from the linear unmixing step are significantly less than one, this will indicate the inability of the set of endmembers to fully explain the spectral signature of the hyperspectral data point. The linear unmixing algorithm chosen was performed through R using the unmix() function in the package RStoolbox [33,42]. It was chosen as the model implies sparsity within the pixel of certain endmembers, meaning that some endmembers within a hyperspectral pixel can be set to zero. This is important as not all endmembers will necessarily be present in all pixels. NNLS unmixing is also widely used in the field of marine studies due to its simplicity and proven ability to yield more accurate results than unconstrained unmixing [43,44]. In addition, NNLS unmixing also decreases fractional retrieval error over unconstrained methods, especially in waters under 5 m of depth, which was the case for this study's dataset. Previous studies have demonstrated that the highest accuracy of classification occurs when the fractional percentage of the endmembers cover over 25% of the pixel recorded [45]. As coral colonies on Heron Reef can span over an area larger than one pixel (>5m wide), accurately determining live coral cover using this method is likely.

### 2.4. Accuracy Assessment

Using RGB/multispectral drone data collected along the same flight paths, an accuracy assessment was conducted. An RGB image was created by mosaicking images collected along the flight path. Live coral cover was estimated for each RGB image through supervised classification using Google Earth Engine, based on the methods of Bennett et al. [13] yielding high classification accuracy of over 85% for live coral. The workflow in this study was modified to suit the format of the dataset and to calculate substrate cover for point sizes comparable to those obtained by drone spectroscopy (Figure 3).

Within the multispectral images, polygons delimiting each substrate class were created to train the classification. The same number of random points across substrate classes were then selected within these polygons to ensure equal sampling and validation of the training data. The Classification and Regression Tree (CART) algorithm was chosen, as the most suitable when compared to Random Forest [46]. To calculate live coral cover, 50 randomly generated corresponding points of overlapping coordinates with the hyperspectral drone data were marked. A pentagonal area of 2.6 meter radius was then demarcated for each point and the live coral cover within each area was calculated. This radius was chosen to equate the circular area of the spectral point's 5.2 meter diameter. Note that in Google Earth Engine the creation of completely circular areas is not possible and therefore the use of a pentagon was used to get the closest equivalent of linear unmixing results. The accuracy of live coral cover assessment through spectral unmixing was then assessed using a Spearman's correlation test between the measured live coral cover (recorded from the RGB classification) and the percentage values obtained from spectral unmixing. This was also conducted for the substrate classes of algae, rock, and sand. Through conducting the linear spectral unmixing, the root mean square error was also obtained for each endmember for an additional measure of error for each individual endmember.

## 3. Results

By combining drone spectroscopy data and a public spectral library, linear unmixing of the spectroscopy points collected on the drone flight was achieved. Over 82% of the spectral variance seen in the drone spectroscopy dataset was explained by the chosen endmembers. With statistically significant correlations between live coral, rock, and sand cover derived from the linear unmixing and the RGB classifications, we highlight the potential for using drone spectroscopy in mapping coral reef habitats.

### 3.1. Evaluating Spectral Libraries

The PCA was conducted on a total of 101 spectra from a public spectral library that were divided into eight substrate classes. There was a lack of distinct clusters for all substrate classes, but with most coral spectra forming a cluster with low scores in the first principal component (Figure 4a). Spectra '56' was chosen as it was projected furthest away from the highest density of algae spectra, with high scores in the first component. The coral spectra chosen was of an Acropora colony, which was deemed appropriate due to the common nature of Acropora in shallow, lagoonal waters, but also specifically at Heron Reef in the area of the data capture [47]. The projections of algae spectra also led to the choice of spectrum '67', which was that of turf algae. This was also deemed appropriate due to turf algae generally being the most abundant algal assemblage found on coral reefs [48,49]. For both sand and bare rock, due to the low number of spectra present in the public spectral library, spectra '80' and '48', respectively were chosen, being positioned away from the other chosen spectra.

The four spectra were normalised, smoothed, and tested for linear dependence, for which results indicated a lack thereof. Spectral signature shapes were compared to known endmember spectra in the literature to validate the likeliness to "pure" spectra. Comparison to the spectra published by Joyce and Phinn [43] confirmed that spectra chosen for the endmember library were suitable (Figure 4b).

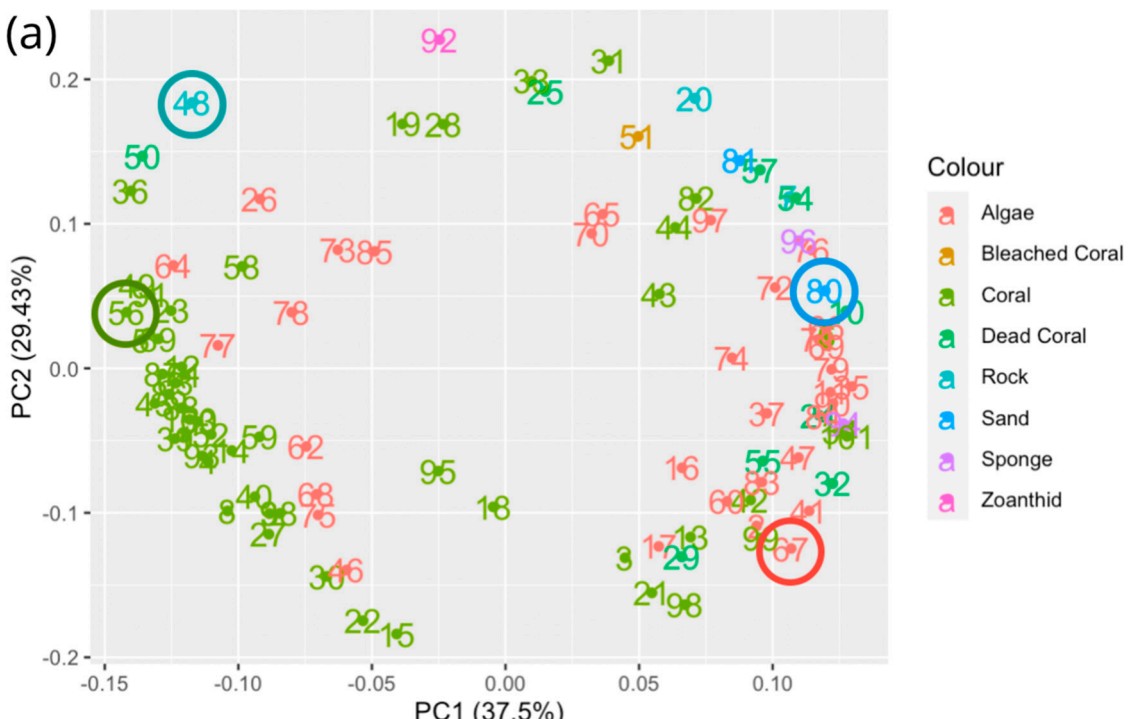

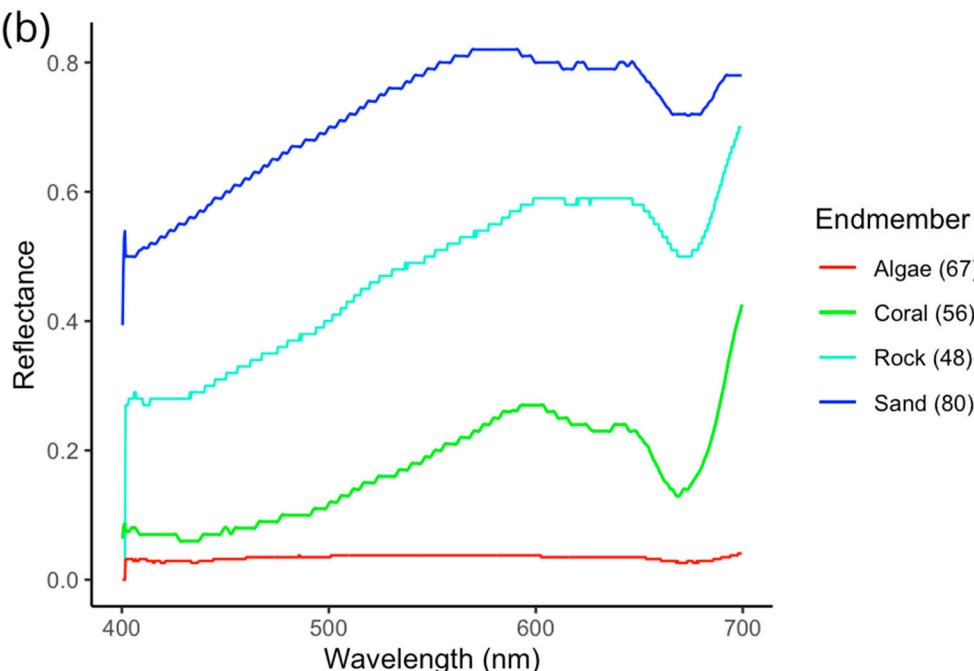

**Figure 4.** (**a**) Principal component analysis of endmember spectral signatures. Chosen endmembers are circled in green (coral), red (algae), green-blue (rock), and blue (sand). Various other substrate classes are also included from the spectral library but were not included in the formation of the final endmember library. The first principal component accounts for 37.5% of the variation, whilst the second component accounts for 29.43%, indicating a lack of full explanation of variance by the first two principal components. (**b**) Spectral signatures of chosen endmembers. Spectra were all smoothed using Savitzky–Golay smoothing and normalised for vector length.

### 3.2. Evaluating Drone Spectroscopy

To challenge the use of public libraries, a PCA was conducted on the drone spectroscopy data to evaluate the potential for endmember extraction within the dataset. As seen in Figure 5i, no clear clusters can be seen, but points were projected across the plot in three directions (a, b, and c). Points projected around "b" and "c" were, respectively situated with low and high scores in the first principal component, whereas points around "a" were projected with high scores in the first and second principal components (Figure 5i). Situating these spectra on a map indicated these represent deep water, coral, and sand (Figure 5ii). This was validated upon further inspection of the spectral signatures, with the deep-water signature showing a characteristic continuous dip in reflectance past 750nm (Figure 5iii). However, due to the spatial resolution of the drone data (circular area of 5.2m diameter) and the heterogeneous nature of coral reefs, it was unlikely that the extracted spectra were as "pure" as those obtained from the public spectral library. The difference in spectral signature between the extracted spectra and the public spectral library spectra could also be attributed to endmember heterogeneity, where the extracted endmembers for coral could have represented different species or even bleached corals. Despite some clustering in the plot, it was difficult to confidently extract "pure" algae and rock endmembers, thus reinforcing the advantages of using the public spectral library.

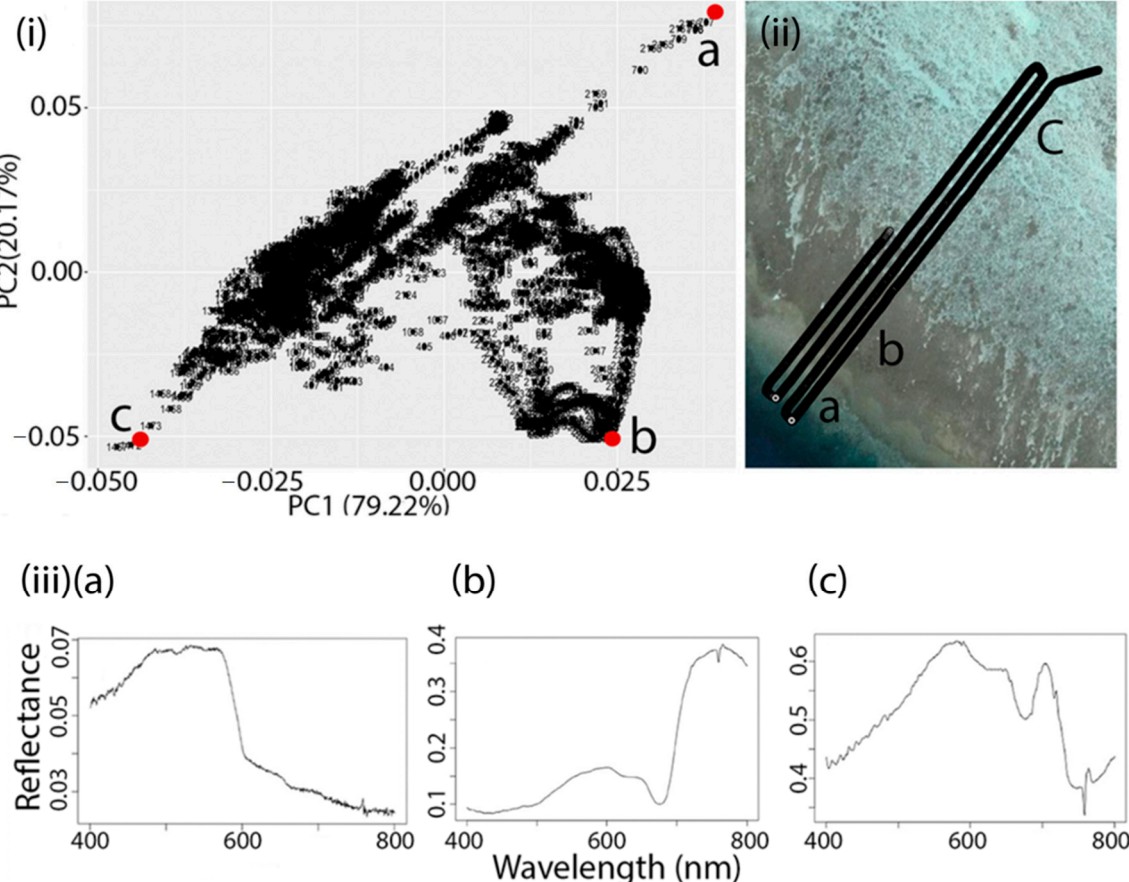

**Figure 5.** (**i**) Principal component analysis of hyperspectral drone data, (**ii**) Map of drone flight, (**iii**) Spectral signatures of self-sampled spectra. Spectra were chosen from the points that clustered the furthest apart, where "a" is likely to represent deep water, "b" coral, and "c" sand. Note that spectra are unlikely to be pure but serve as the purest spectra within the drone dataset. Axes are not shown to the same scale for better visualisation of spectral trends.

### 3.3. Unmixing Drone Spectroscopy

A total of 2,553 reflectance measurements were unmixed during the spectral unmixing step using the selected endmember library. Spectral unmixing of the drone data using the endmember library created revealed a live coral coverage ranging from 0 to 24% across the drone flight path studied. An increasing coral cover gradient can be observed progressing away from the island (Figure 6a). Similarly, rock cover decreased along the same gradient, but was found in lower density compared to live coral, ranging from 0 to 17% (Figure 6b). Conversely, sand cover is higher on the sandy reef areas with sand cover ranging from 0 to 64%, as expected. Data points where no sand influenced the spectral signatures all coincide with the highly structured section of the reef preceding the reef slope (Figure 6c). Algae showed the greatest range of percentage cover, of 0 to 69%. As seen on Figure 7d, most points displayed a percentage algal cover between 25 and 60%, which is high compared to the other substrate classes.

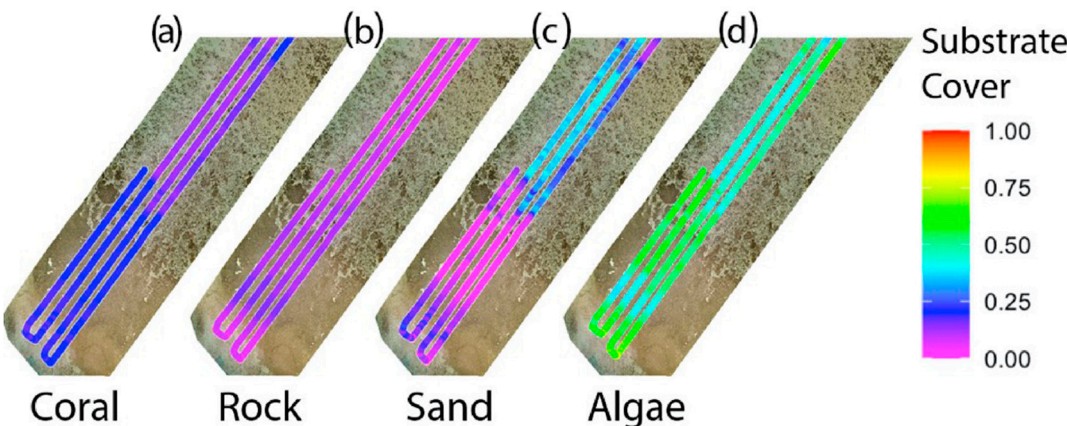

**Figure 6.** (**a**) Percentage benthic habitat type estimated from linear unmixing using drone spectroscopy: (**a**) Coral, (**b**) rock, (**c**) sand, and (**d**) algae. Results were overlaid on a map of the study area in question. Substrate cover is shown from a scale of 0 to 1. The model yielded an RMSE of 0.00204.

As fractional contributions of endmembers were not forced to sum to one (100%), the unexplained fractional contributions may be explained by endmembers that were not included in the endmember library, such as species within the same class with variable spectra or completely separate substrate classes such as marine biota or mud. Overall, the summed percentage cover of the four endmembers for all points ranged from 82 to 100%, showing that the endmembers chosen were able to account for at least 82% of the spectral mixing seen within each drone point. Over 78% of data points studied showed total percentages of over 90%. This demonstrates that the use of only four endmembers can produce a relatively representative map.

### 3.4. Accuracy Assessment

To check the validity of the results, an accuracy assessment was conducted to compare the unmixing results to a classification of fifty polygonal areas (Figure 7). As seen on the classified mosaicked image, the inner reef flat showed the greatest number of pixels being classified as sand. Further towards the crest, algae is the dominant substrate class, with coral and rock substrate types increasing in this area as well (Figure 7).

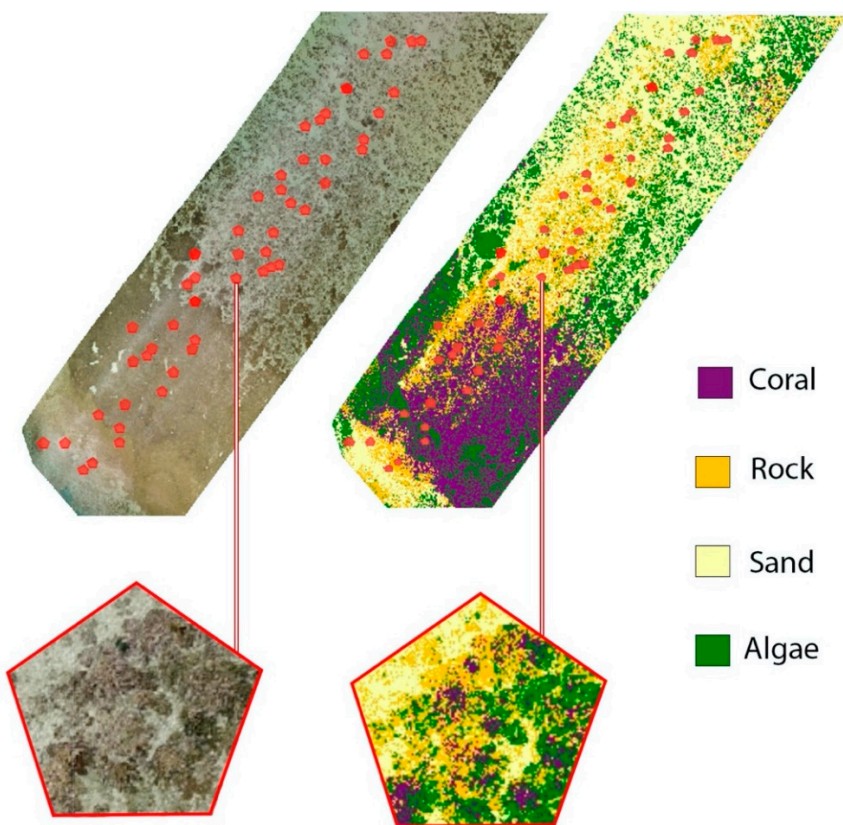

**Figure 7.** Mosaicked RGB image of the corresponding study region showing the fifty randomly generated pentagons to calculate substrate cover with a classified map of the four substrate classes: coral (purple), rock (orange), sand (yellow), and algae (green).

Retrieving fractional contributions by linear unmixing revealed an unsurprising spatial distribution of the endmembers unmixed. Spearman's correlation tests revealed a significant moderate correlation between live coral cover derived from spectral unmixing and from RGB classification (rs = 0.408, S = 13085, p = 0.00297) (Table 1). Results estimated a mean live coral cover of 17% and 14%, respectively for the unmixing and RGB classifications. As seen on Figure 8a, this was predominantly the case at low to moderate coral cover (Figure 8a). Although a correlation is seen, in order to better test the correlation between the unmixing results and the RGB classification, a greater range in coral cover would need to be tested.

Similar to live coral, rock cover yielded a moderate correlation between classification results (rs = 0.505, S = 10943, p = 0.000158). However, rock cover was underestimated in the linear unmixing process when compared to the RGB classification (Figure 8b). This underestimation may have been the result of misclassification within the accuracy assessment, by falsely classifying other benthic types as rock. Confusion between rock and algae is especially likely due to the difficulties in distinguishing turf algae that may be overgrown on rock or dead coral specimens. This would have resulted in an overestimation of rock in the RGB classification.

On the other hand, algae cover was shown to have a low and insignificant correlation between unmixing and RGB classification results (rs = 0.115, S = 19570, p = 0.424) (Figure 8c). Again, this may be due the inability of distinguishing between turf algae and other benthic groups in the RGB classification, but could also be linked to human error in the training step, being limited by less spectral information and inefficient spatial resolution to confidently classify groups.

Sand classified by linear unmixing had the highest correlation with that obtained from the accuracy assessment, likely meaning that the sand measured is in truth, sand (rs

= 0.620, S = 8392.8, p = 1.208 × 10-6). Sand was underestimated in the linear unmixing process, which could potentially be explained by error in the unmixing step, but also could be attributed to misclassification in the RGB classification (Figure 8d). Sand could have been underrepresented due to the unmixing algorithm detecting spectral influences from other substrate classes such as algae and small biota that may be too small to be visualised with the resolution available from the RGB image. Sand may also be variable in origin, grain size, and mineralogy, and therefore one endmember may not explain the spectral mixing caused by both silicate sand and carbonate sand from bioeroders such as parrot-fish and physical erosion.

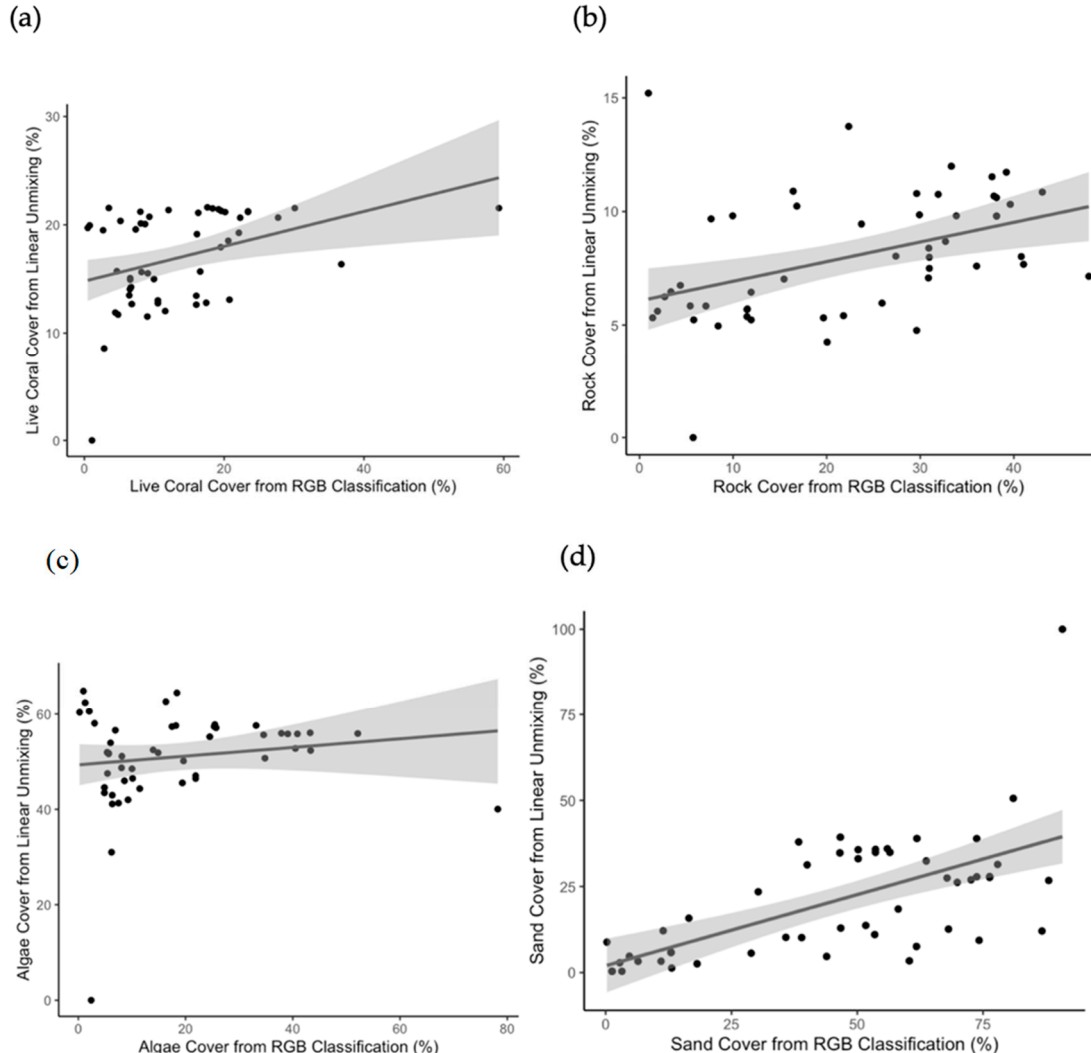

**Figure 8.** Relationship between substrate cover determined from linear unmixing and from the RGB classification. The black line represents the best fit line and the grey area indicates the 95% confidence interval. (**a**) Regression results for live coral cover, (**b**) Rock cover, (**c**) Algae cover, and (**d**) Sand cover. Axes are not shown to the same scale for better visualisation of individual trends.

**Table 1.** Results of Spearman's correlation test. Significant correlations are highlighted in bold.

| Substrate type | $r_s$ | S | p |
|---|---|---|---|
| Live Coral | 0.408 | 13085 | **0.00297** |
| Algae | 0.115 | 19570 | 0.424 |
| Sand | 0.620 | 8392.8 | **1.208 × 10⁻⁶** |
| Rock | 0.505 | 10943 | **0.000158** |

## 4. Discussion

### 4.1. Using public Spectral Libraries

Successfully using public spectral libraries for these types of analyses encourages mapping efforts by making unmixing studies more accessible (by decreasing the need for field collection of endmembers), as well as opening doors for the facilitation of endmember determination [28]. It is important to note that determining endmembers arguably remains the most crucial step in the spectral unmixing process [50]. Acting as the first step along with the pre-processing of data, minimising error is vital, as error caused from insufficient or unrepresentative endmember selection can propagate errors in all subsequent steps of the analysis.

Despite yielding positive unmixing results, direct improvements to the methods can be achieved in the future through additional selective steps. Choosing endmembers from the PCA plot is the only step that is not automated in the linear unmixing workflow and relies on user choice. Although the endmembers chosen were successful in unmixing the drone spectroscopy points in this study, the methods used in choosing endmembers should be automated to remove bias and ensure repeatability. The method is also flawed in that choosing endmembers that cluster far apart on a PCA plot could lead to the extraction of anomalies, leading to the use of endmembers that are not representative of their substrate class. This stresses the need for standardised endmember determination methods. Examples of methods include endmember determination include iterative endmember selection (IES) or endmember average RMSE (EAR) [50]. Using automated steps such as these will help in standardising the proposed workflow of this study and ensure that the choice of suitable endmembers is statistically backed.

With marine public spectral libraries becoming more accessible and complete, it may be soon possible to find pure spectra with matching sensors and environmental conditions as those collected for individual studies, facilitating the pre-processing step by decreasing the need for extensive normalisation of datasets. In order to achieve this, future data collection methods should aim to standardise methods for collecting spectra for libraries and provide additional information on the factors affecting intra-specific variability, such as developmental stage, tidal position, and bathymetric position [51]. Public databases are commonly used in the fields of mineral exploration and canopy analysis, where organisation and individual researchers have combined efforts to develop shared libraries for a range of different materials, both natural and anthropogenic. To minimise the variation in spectra caused by differences in data collection techniques, various standardisation methods have been proposed, such as continuous wavelet analysis, a form of scaling spectra [52]. By doing this, spectral libraries become increasingly transferrable between studies and the use of spectra from different libraries can be made possible. Shared public databases such as USGS, SPECMIN, and SPECCHIO also help in identifying the requirements of a spectral library, by using a Database Management System (DBMS) that stores spectral information in relational tables [53]. However, this does not necessarily enforce data integrity, reinforcing the need for standardisation methods during data collection.

### 4.2. Benthic Distribution on Heron Reef

According to the current Reef Check Australia Health Report of Heron Island, the reef comprises of approximately 37% live coral, similar to the 36% in 2017, the year the drone spectroscopy data were collected [54]. In that year, across 17 sites studied, hard coral cover ranged between 3% and 65%, which is the range within which the unmixed fractional contributions fell within. The highest coral cover was highest at the reef slopes and the lowest on sandy reef flats, which agreed with findings by the Reef Check Report [54]. Although the unmixing results fall within the live coral cover range found by Reef Check, comparing results to monitoring studies must always be done with caution, as these in situ studies often overestimate live coral. This is often the case due to bias in choosing monitoring sites, where the sandy regions tend to be monitored less frequently.

Additionally, the findings of this study were based on one single drone flight and therefore may not serve as an accurate representation for the benthic distribution on the rest of Heron Reef. This could explain the slightly lower overall coral coverage yielded by the unmixing at 17% compared to the estimated 37% found by Reef Check.

For algae, a previous study by Roelfsema et al. [55] found that chlorophyll a concentrations found in Heron Reef sediments were among the highest reported for any marine sediments. This was especially the case on the windward side of the reef, which is where the drone data from this study was captured. The sediments sampled were used to quantify benthic microalgal communities [46]. The high levels of benthic microalgae could be a factor explaining the dominance of algae seen in the findings of the spectral unmixing, as the endmember of algae could have extracted the fractional contributions of turf, macro- and microalgae combined. Similarly, this could also explain the low rock cover found through the unmixing process, as rock covered by turf algae is likely to have a spectral signature similar to that of the turf algae endmember used.

### 4.3. Sources of Error and Potential Improvements

Weak correlations in the accuracy assessment may be attributed to error in the data collection and error in misclassification during data processing. Whilst errors in the data processing generally occur in the quantitative analysis and classification steps, errors in data collection are accumulated through the sensors and platforms, the classification steps, and the water column, as previously mentioned [56]. As this study involved combining three separate datasets, errors produced within collecting or processing of all three need to be considered.

#### 4.3.1. Sources of Error from Sensors and Platforms

As discussed, sources of error from sensors and platforms may arise due to the instability of the moving drone platform and inaccuracies in geopositioning. To avoid this, spectrometers may be attached on a gimbal. However, not all commercial-grade drones have a built-in gimbal and attaching one will add additional weight and cost. Errors in geopositioning present implications for the accuracy assessment step, as matching up the coordinates between the drone spectroscopy data and the mosaicked RGB image will not be exact. This could explain the lack of correlation seen in the accuracy assessment, as spatial inaccuracy, even minimal, can lead to significant changes in substrate cover in a heterogeneous environment. As the spectroscopy data are not in the form of imagery and do not provide spatial context, matching up of data through landmark structures is not possible. This highlights one of the drawbacks of this study's chosen accuracy assessment.

#### 4.3.2. Sources of Error from the Classification Steps

It must be noted that the accuracy assessment used in this study serves as one option to testing accuracy without the need for underwater data collection. The RGB classification in itself presents inaccuracies, as it relies on visual classification by the user and is therefore prone to human error and bias. It is also limited by the amount of spectral information it holds and is more likely to confuse benthic groups such as coral and algae [27]. Therefore, the classification obtained from the linear unmixing has the potential to show higher accuracy compared to that obtained from the RGB image. To improve the RGB classification accuracy, more polygons and points could have been used to train the classifier and sun glint could be added as a substrate group to minimise misclassification. In making sure that we collect data in the most appropriate way in the first place, we minimize artefacts due to sampling and environmental conditions [57]. As this study establishes a protocol where in situ underwater validation was not conducted as part of it, further testing is recommended for future studies to validate this.

As previously discussed, GPS location errors also arise during underwater validation and images collected underwater in situ cannot be compared at the same scale. Efforts to

minimise GPS location errors include the use of georeferenced quadrat sampling in estimating benthic cover, combined to underwater photography [13]. However, this would greatly increase data collection effort and does not address the issue of scale. An alternative would be to conduct an accuracy assessment using imagery collected from the same drone and at the same time of drone spectroscopy collection. This would reduce spatial discrepancies between the spectroscopy dataset and the validation data, as both would be collected from the same source. Although this may serve as a credible accuracy assessment, the need to develop more effective methods for validation is highlighted.

Aside from the accuracy assessment, misclassification errors could have occurred in the linear unmixing step. These errors could be linked to inefficient data reduction, the absence of representative endmembers, or the confounding presence of the water column. Studies have found that many of the differences between coral and algae lie between 520 and 580nm and therefore linear unmixing could have been conducted on a dataset where these wavelengths were given a greater weighting [13]. Hochberg et al. [27] used a multivariate stepwise selection procedure to isolate the wavelengths that best differentiate between substrate classes. Spectral feature selection is another method that relies on extracting endmembers that minimise intra-class variability and maximise inter-class variability [58]. These methods remove less meaningful information in the dataset for more efficient classification. Inefficient data reduction could therefore be improved by focusing on wavelengths where diagnostic features of substrate classes can be found, but the disadvantages of losing data must be considered.

Inaccurate estimation of benthic cover could have also occurred in the linear unmixing step by not including certain endmember classes (biota such as holothurians) or not accounting for endmember variability within the analysis [59]. Due to the inherent spectral variation that occurs within and between species of the same class, using one endmember spectra per substrate class leads to an oversimplification of the model that does not incorporate the heterogeneous nature of coral reef habitats [50]. Algae comes in the form of more than turf, with various species of red, brown, green, fleshy and calcareous algae, whereas corals can be classified as bleached, blue, brown or soft/gorgonians, that each differ in spectral signatures [60]. In order to account for spectral variability within endmember classes, previous studies have used averages of various species and yielded a lower overall RMSE.

Alternatively, implementing MESMA instead of single endmember SMA accounts for endmember variability [45,61]. Using MESMA, where different endmember spectra can be chosen depending on the pixel, has shown to yield lower RMSE values in coral reef unmixing studies in the past [62,63]. However, MESMA can also be flawed as it cannot fully incorporate the heterogeneous nature of coral reefs, only choosing one endmember spectra per class in each pixel or point [62]. To evaluate its potential with point spectroscopy data such as that used in this study, further research should be conducted. Results can then be compared to those from studies in which endmember variability is not accounted for, or where an average signature is used to represent one endmember class.

There are significant biological implications associated with grouping species together within endmember classes or omitting substrate classes. Although some mapping studies may not require the differentiation within algal and coral groups, the use of such proxies for coral reef health could be misleading. Generally, an increase in turf or macroalgae over time can represent a phase shift from coral-dominated to algae-dominated reefs, indicating a decline in reef health as the presence of some algae affects coral recruitment and survival [48]. However, observed increases in crustose coralline algae (a red alga) can instead be an indicator of increasing coral reef health [48,64]. Although this study is a development and test of a workflow that does not include testing biology, it is important to consider for future applications of the technique what biological implications the dataset being used can have.

4.3.3. Sources of Error from the Environment

This study was conducted on a shallow study site, when environmental conditions were good and water quality was high, and therefore it was an optimal study site to test the effectiveness of the workflow in ideal conditions. If this method were to be applied in deeper water, water column correction would be needed. Lee et al. [25]'s algorithm consistently showed improvements in classification accuracy when applied. This should be used to build upon the workflow in this paper, for future studies requiring water column correction. Combining a semi-analytical model with linear unmixing on hyperspectral imagery has been achieved with positive results by Goodman and Ustin [24] and Klonowski et al. [65], demonstrating the potential for using such models on spectroscopy data. It is important to note that water column correction remains a difficult task, explaining the choice to exclude it from this study, for the sake of simplicity in using it for shallow reefs. Nonetheless, through the use of linear unmixing techniques, this workflow serves as a first step towards scaling mapping of hyperspectral point data and can be added to in order to incorporate water column correction.

### 4.4. Examples of Future Applications

Although accurate measurements have been done using the relatively more affordable RGB data, there are benefits of using data with a greater amount of information. Mapping studies, such as that by Bennett et al. [13] focused on using RGB images to extract substrate cover, instead of spectroscopy, and showed the pros and cons of doing so. Although yielding positive results, the paper highlighted the difficulties in differentiating between certain substrate types such as live coral and rock, where live coral cover estimates are often overestimated due to rock being classified as coral. The reason for the use of spectroscopy in this project was to assess whether the use of a greater number of wavebands within the data would help to differentiate between similar looking substrate classes. In the case of estimating live coral cover, or the cover of other types of substrata, a 1D coverage could be an effective way to obtain estimates whilst ensuring a higher accuracy of classification than provided through the use of RGB images. Using such data can be useful for more sophisticated information extraction purposes in the future.

Although spectroscopy has been shown to successfully help in monitoring live coral cover, it is not limited by this application. Since it provides a way to access complex datasets without the need for extensive expertise in remote sensing, the proposed workflow could be used in various fields such as quantitative mapping, through monitoring bleaching and reef health, without being restricted by the environmental and time limitations offered by a satellite. Joyce and Phinn [66] used hyperspectral imagery to derive chlorophyll content of coral reef substrates. Quantifying pigment concentrations using drones may serve as early warnings for bleaching or health monitoring on the reef for conservation managers. Drone spectroscopy could be further applied to quantitative mapping by quantifying in situ fluorescence spectra of benthic substrates, which if further tested, could open doors to quantifying photosynthetic potential of the substrata [67]. This gives an indication of applications of drone spectroscopy that need to be tested, which could facilitate monitoring through directly quantifying key variables. Developing this workflow for mapping substrate cover demonstrated a relatively simple application, but helps to present a method that enables a range of other more sophisticated applications. The applications are endless and the simplicity of running the code makes these applications achievable.

### 5. Conclusions

Overall, using drone spectroscopy data shows promise for mapping benthic cover on Heron Reef. This type of data offers a low-cost resource that has the potential to provide a level of detail and information that cannot be provided by multispectral and RGB data.

The process of determining endmembers in this study was able to account for over 82% of the spectral mixing throughout all spectral measurements collected from a con-

sumer-grade drone and was able to moderately determine the exact fractional contributions of live coral, sand, and rock. Although there still remains the need to further refine current workflows, this method provides an accessible process that can be applied to data collected by affordable technology. Due to this, future research should focus on testing the effectiveness of using drone spectroscopy for specific applications, such as quantitative mapping or detecting coral bleaching. Further recommended steps to improve the study include an automated endmember selection step, bathymetric retrieval, and water column correction.

This highlights the importance of this study, as it can hopefully help further widescale research and monitoring programs, not only in highly studied sites, but in remote areas. With the increase in accessibility to both drone hyperspectral data and public spectral libraries, high spectral resolution information will be made available for mapping studies for a range of various research, as the applications for remote sensing are endless.

**Author Contribution**s: Conceptualization: V.J.C. and K.E.J.; methodology, V.J.C.; data collection, K.E.J.; formal analysis, V.J.C.; original draft preparation, V.J.C.; writing—review and editing, V.J.C. and K.E.J. All authors have read and agreed to the published version of the manuscript.

**Funding:** Internal JCU staff grants to Dr Karen E. Joyce and Dr Stephanie Duce provided funding for field survey and data acquisition. There was no external funding provided for this project.

**Data Availability Statement:** The spectral endmember library collected by Dr Christian Roelfsema and Dr Stuart Phinn is openly available in Pangaea at https://doi.org/10.1594/PANGAEA.804589 (accessed on 3 Oct 2020), reference number 804589. The RGB images that were used to create the mosaicked image for accuracy assessment can be found at https://data.geonadir.com/project-details/173 (accessed on 1 April 2021). R Script used in RStudio for linear unmixing of hyperspectral points is available online at https://github.com/valeriecornet/DroneSpectroscopy/blob/main/R%20Linear%20Unmixing.R (accessed on 1 April 2021). JavaScript code used in Google Earth Engine for classification of the RGB mosaicked image is available online at https://github.com/valeriecornet/DroneSpectroscopy/blob/main/RGBClassification (accessed on 1 April 2021). Script was modified from Bennett et al. (2020)'s code and added to. RGB drone data is available via https://data.geonadir.com/project-details/173 (accessed on 1 April 2021).

**Acknowledgments:** We thank Stephanie Duce for assistance with drone data collection and Arnold Dekker for his valuable input on the study and for pointing towards the spectral endmember library used. We thank Christian Roelfsema and Stuart Phinn for sharing their valuable data. We thank Katie Bennett and Florence Sefton for sharing the JavaScript code that was used and modified for the classification of the mosaicked RGB image. Finally, thank you to Jonathan Kok, Raf Rashid, Redbird Ferguson, and Joan Li who reviewed and provided useful comments on drafts of the paper.

**Conflicts of Interest:** The authors declare no conflict of interest.

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
