# Peer review of "Assessing the Potential of Remotely-Sensed Drone Spectroscopy to Determine Live Coral Cover on Heron Reef"

_drones, doi:10.3390/drones5020029_

Round 1

Reviewer 1 Report

In the manuscript was conducted a relevant study quantifies the amount of various benthic substrates using drone-based spectroscopy on Heron Reef.

This is a well-prepared manuscript.

The application of this approach is in the field of the journal.

There are minor editorial and grammatical corrections needed throughout the text.

Authors can improve the quality of the manuscript by explaining better the motivation and importance of the research in the introduction.

Should better emphasize the future research of the authors in the conclusions.

Reviewer 2 Report

It is a very interesting and well-designed work. 
The methods and materials were very well explored, and the results corroborate the proposed approach.

The only major problem is that, although interesting, the concept is not new, and a more in-depth study about different approaches should be carried on. For instance, there are a lot of better approaches than PCA, so why to use it without propoer justification?

One way to improve the present work's value is, if possible, to upload the data through a data server such data port or similar that provides a DOI.

Minor issues;
provide a better workflow, adding the outputs of each step. (with a formal mathematical presentation.)

Reviewer 3 Report

The authors tested the ability of using drone spectroscopy to determine substrate cover through spectral unmixing on a portion of Heron Reef, Australia. As discriminating between substrate types has previously been achieved with in situ spectroscopy but has not been tested using drones, so this research is interesting and important. The paper is also well written, so I suggest to accept it if the authors can answer the following minor questions:

  1. Line 96-98: Representative spectra were chosen from the public spectral library of substrata collected in situ on Heron Island in 2006 and provided by Dr Arnold Dekker [20]. I suggest the authors provide a web link in the text so that the readers can download the public spectral library to repeat the experiments.
  2. Line 181-186: Due to the time limitations presented by the study and the aim of shaping a more accessible, repeatable, and relatively simple workflow, water column was not corrected for using radiative transfer equations. Previous studies have confirmed that classification of reef substrata using the aforementioned spectral range remains possible at depths shallower than six meters, which was the case for this study [31]. The authors may not be able to provide some results that consider the water column correction, but I suggest the authors to provide some possible ways to correct the influences of the water column, and what will happen when the water depth is greater than six meters.
  3. Figure 4 and Figure 5: The shapes of the spectral signatures of self-sampled spectra (from the drone spectroscopy data) seem to be very different from corresponding shapes of the spectral signatures of chosen endmembers (from the public spectral library). Why it is reasonable and effective to use the spectral signatures derived from the public spectral library in the unmixing approach?
  4. The most important question is: the authors utilize the RGB images to assess the substrate cover results of the drone spectroscopy data, why the authors did not utilize the RGB images to extract the substrate cover? Traditionally, a camera that can capture a RGB image is usually cheaper than a drone spectroscopy equipment. Furthermore, RGB images can provide detail 2D distribution information of the substrates, while the drone spectroscopy equipment can only provide 1D information.

Reviewer 4 Report

The manuscript discusses the potential use of spectroscopy drone for assessing coral reef live cover on Heron Island. The subject is interesting and useful, however there are some comments and suggestions that need to be considered.

  • The authors need to highlight the aim of study and it's background. There is also some need for re-organization in the manuscript.
  • In the abstract the authors say that spectroscopy using drones hasn't done before, and their study uniquely showcases this. I do not think this is correct. The drone remote sensing-spectrometry has been around for some time. One example even mentioned by authors at the end of introduction. 
  • Lines, 66-72, authors say "little work has been done to test the extent to which the more affordable point based spectrometers" this should reflect in the title, abstract or introduction. 
  • There is no in situ data provided for the accuracy assessment. The authors used RGB images to compare and also used public spectral data for their study. My question would be why all these hasn't been checked against the actual substrate in the field ? 
  • Section 4.2 can go into material and method, and introduction section.
  • Other references and literature should be considered in the introduction and discussion. 
  • Section 4.3, I suggest to decrease this and move part of it into introduction. Please be specific about the errors in your study. The current text is too long. I would summarize all in one title, rather than several subtitles. Remove anything that is no specifically related to your study. 
  • same also for 4.4 
  • Conclusion, please be specific. Just summarize your findings, and out comes. 
